# Self-Consistent Explanation of the Untwist Alignment of Ferroelectric Nematic Liquid Crystals with Decreasing Cell Thickness and Deviation of the Surface Easy Axis Experimented upon Using the Brewster Angle Reflection Method

**Sakunosuke Abe** [iD] **, Yosei Shibata, Munehiro Kimura** *[iD] **and Tadashi Akahane**

Graduate School of Engineering, Nagaoka University of Technology, 1603-1 Kamitomioka, Nagaoka 940-2188, Japan; s203109@stn.nagaokaut.ac.jp (S.A.); y_shibata@vos.nagaokaut.ac.jp (Y.S.); tadashi-akahane@outlook.jp (T.A.)

\* Correspondence: nutkim@vos.nagaokaut.ac.jp; Tel.: +81-258-47-9540

**Abstract:** The huge dielectric constant of ferroelectric nematic liquid crystals (FNLCs) seems to bring about a difficulty of molecular alignment control in exchange for a potential device application. To obtain a satisfactory level of uniform molecular alignment, it is essential to understand how the molecules near the alignment surface are anchored. In this study, bulk molecular alignment with an anti-parallel rubbing manner, which has not yet been investigated extensively, is explained using a conventional torque balance model introducing a polar anchoring function, and it is shown that the disappearance of the bulk twist alignment with decreasing cell thickness can be explained self-consistently. To validate this estimation for a room-temperature FNLC substance, the Brewster angle reflection method was attempted to confirm the surface director's deviation from the rubbing direction caused by the polar surface anchoring.

**Keywords:** ferroelectric nematic liquid crystal; surface polar anchoring; elastic theory; rubbing; Brewster angle reflection method; polarization optical microscope; differential scanning calorimetry





## 1. Introduction

The intrinsic properties of nematic liquid crystals (NLCs), such as optical anisotropy and changes in molecular alignment in electric fields, have attracted much attention for their advanced display and optical applications. Born et al. predicted the presence of NLCs exhibiting intrinsic polarity in 1916 [1]. This theoretical speculation went largely unheralded for a century before being unequivocally validated with the discovery of ferroelectric NLCs (FNLCs) in 2020 [2]. FNLCs are characterized as fluids that exhibit a macroscopic electric polarization $p$, which is distinguished by a molecular arrangement from an ordinary nematic phase [3–5]. A notable property of these LCs is that the direction of the director (denoted as $n$) does not coincide with the spontaneous polarization, $p$ [6] and thus FNLCs manifest physical properties that diverge from ordinary NLCs. This peculiarity means that $n$ is not symmetrically equivalent to $-n$, signifying a perturbation in head-to-tail symmetry [7]. The inherent characteristics of asymmetric molecules in FNLCs have been reported to induce phenomena that create a high polar order with a huge dielectric constant [3,6]. In general, most materials that have huge dielectric constants possess specific low symmetries within their atomic arrangement. The emergence of FNLCs challenges this classical understanding, thereby eliciting profound interest in FNLCs from the viewpoints of both material physics and the development of novel applications [8,9]. Recently, the first demonstration of converse piezoelectricity using FNLC substances that exhibit significant ferroelectricity at room temperature was published [10,11], and these results pave the way to the realization of flexible actuators that cannot be realized with solid materials. Studies on the unique electrical attributes of FNLCs have demonstrated

that they can be distinguished from ordinary LCs using methods such as temperature-dependent dielectric constant measurements [6,12] and the detection of second harmonic generation [7,13]. Additionally, it has been determined that multifarious polar alignment structures form within bulk samples, which are driven by interactions between the substrate surface and NLC molecules [14,15]. Furthermore, intermolecular interactions along the in-plane direction induce surface polarity in FNLCs. Recent studies have revealed that thin slabs filled with FNLC molecules form the stable characteristic polar single domains, while thicker slabs exhibit certain complicated formations of multi-domains [16]. These findings have been predominantly derived from the observations of color changes using polarization optical microscopy (POM) and the measurement of twist angles on the bulk [15,17,18]. Fluorescence confocal polarizing microscopy is a powerful tool for obtaining a three-dimensional molecular alignment distribution or topological textures [19]. Up to today, intensive research on the bisecting properties of conic sections has been demonstrated by making full use of the analytical geometry [20].

When ferroelectric smectic liquid crystals were vigorously studied in the 1980s, obtaining a uniform alignment was one of the issues in developing a liquid crystal display [21]. There was no doubt that a breakthrough in forming monodomains was the invention of the surface stabilization technique [21,22], through which an unwind helix is achieved by reducing the sample cell thickness. It is natural to think that uniform monodomains could be achieved even in FNLCs by reducing cell thickness due to the elastic repulsive force. Certain attempts have been demonstrated [15,16] where the first ferroelectric nematic substances of RM734 and DIO [2–5] were used, respectively. On the one hand, it was reported that a uniform orientation can be achieved through a parallel rubbing manner if the cell thickness is thin [16]. On the other hand, large monodomain patterns for both substances can be achieved despite the lower anchoring strength expected for photoalignment regardless of cell thickness [23]. It is assumed that this is because the polar anchoring induced at the alignment surface is weak in the case of photo-alignment [23] and bi-directional rubbing [15]. Even in an FNLC cell with an antiparallel rubbing manner that tends to form a twist alignment, molecular alignment will be uniform when the cell thickness is thin [15,16]. It would be beneficial if these phenomena could be understood comprehensively using a conventional torque balance model [24] introducing a proper anchoring function.

This research aims to derive a simplified torque balance equation from classical continuum theory and provide a phenomenological explanation with regard to a room-temperature FNLC substance that has recently attracted attention. The validity of this phenomenological explanation is thoroughly demonstrated through empirical evidence, and the torque balance equation is shown to be theoretically sound and to have practical applicability. In studying the complexity of surface interactions, we found that adjusting the cell sample's thickness realizes uniform molecular alignment at the surface under a certain surface anchoring energy. This phenomenon eliminates the intrinsic $\pi$-twist alignment associated with the polar molecular alignment structure, thus giving rise to a planar alignment state. Studies towards obtaining the ferroelectric nematic phase ($N_F$) have focused on two LC materials, DIO and RM734, though in recent years other FNLCs have been reported such as FNLC-919 [9–11,17], which exhibits significant ferroelectricity at room temperature. In this study, we used FNLC-919 as it is easy to handle in experiments without a thermostat. Similarly to DIO and RM734, FNLC-919 shows a typical polar alignment structure and the characteristics of an extraordinary dielectric nematic phase, as reported by Yu et al. [17]. To investigate the validity of our theoretical LC model, we estimated the LC alignment changes for LC molecules at the surface using the Brewster angle reflection method (BAR method) [25]. This method is not affected by phase retardation and is suitable for analyzing thin slab cells. We will also consider the influence of rubbing strength on surface anchoring characteristics.

## 2. Materials and Methods

### 2.1. FNLC Substance and Sample Cell Praparation

The FNLC substance used in this study was FNLC-919 (Merck, Rahway, NJ, USA) [9–11,17]. The original phase sequence is as follows [10]; Iso. 80 °C $N_1$ 44 °C $N_2$ 32 °C $N_F$ 8 °C Cryst. $N_1$ and $N_2$ represent an ordinary nematic phase and an intermediate phase between the $N_F$ and the $N_1$ phase, and these nematic subphases are out of our interest. As described below, it is necessary to dope FNLC-919 with light absorbers to evaluate the molecular alignment in the BAR method. In this study, 4-dimethyl-amino-azobenzene (DAB, Thermo Scientific Co., Ltd., Waltham, MA, USA) was mixed with FNLC-919 with a stirrer to prepare an FNLC mixture exhibiting monochromic absorption. DAB, a famous azo dye, is frequently used in experiments on liquid crystals because of its rod-shaped molecules and high miscibility with liquid crystals. The dosage of the light absorber should be such that enough light absorption can be achieved without changing the phase sequence. UV–Vis absorption measurements (V-550, JASCO Co., Ltd., Tokyo, Japan) were made on the FNLC-919 and DAB mixture, recording a peak wavelength absorption at 416 nm. After some preliminary experiments, the optimum concentration of DAB added to the FNLC-919 was decided to be 5 wt%. Investigations have already been carried out on utilizing the anisotropic extinction coefficients at the wavelength of 419 nm when using 5CB as a host NLC and adding 5 wt% DAB constituting $k_e = 0.064$ and $k_o = 0.018$, respectively [25]. The resultant penetration length for the mixture of FNLC-919 adding 5 wt% DAB will be approximately 1 µm. The phase transition temperature for this mixture was analyzed through differential scanning calorimetry (DSC, DSC7020, Hitachi High-Tech Science Co., Ltd., Tokyo, Japan) to check that $N_F$ still occurs when DAB is added to FNLC-919. Figure 1 shows that the phase transition temperature from the $N_2$ to $N_F$ phase decreases upon dye addition. From these DSC measurements, we decided to carry out the BAR method measurement at 20 °C. It is also confirmed that the photoisomerization reaction of the added DAB is negligible in our experiments under a weak light source such as a halogen lamp.

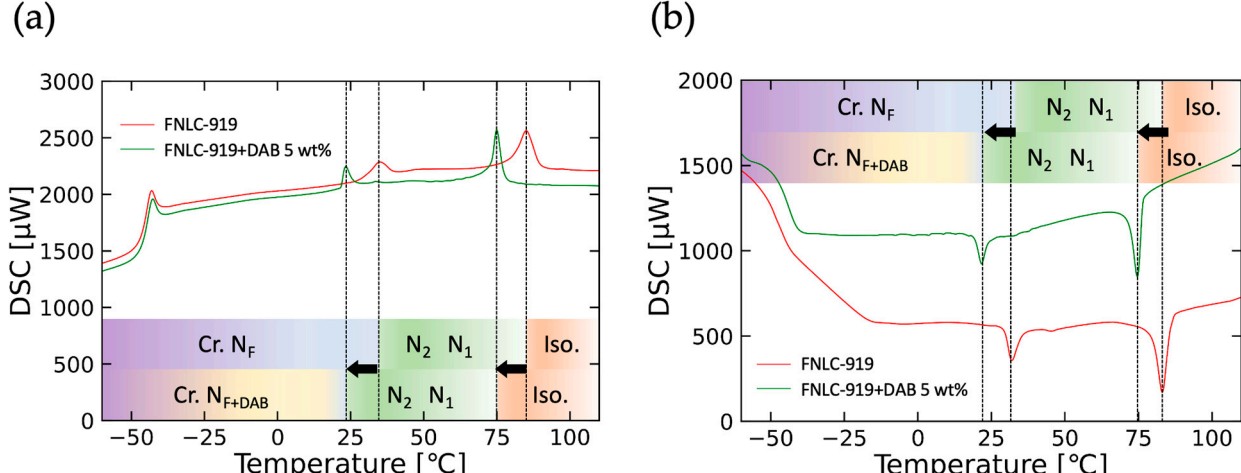

**Figure 1.** Thermal analysis of phase transition temperatures influenced by the addition of DAB to FNLC-919. A DSC assessment was undertaken to determine how a 5 wt% mixture of FNLC-919 and DAB modulates the phase transition relative to pristine FNLC-919. (**a**) Exothermic peaks during the temperature decrease. (**b**) Endothermic peaks during the temperature increase. The respective masses during the measurements were 8.20 mg for the pristine sample and 8.23 mg for the mixture, with a controlled temperature rate of 10 °C/min.

Sandwich-type LC cells were fabricated as follows. First, pre-coated soda–lime glass substrates with indium-tin-oxide (ITO) thin films as charge leakage electrodes were prepared, where the sheet resistance was 50 Ω/sq. Then, a solution of organic solvent and polyimide was spin-coated onto the ITO films, followed by a baking process at 180 °C for

an hour to remove the solvent. The surface of the polyimide film with a thickness of less than 100 nm was then rubbed unidirectionally using equipment (Joyo Engineering Co., Ltd., Tokyo, Japan) as illustrated in Figure 2. Explaining in detail the rubbing process is significant in this subsection since there are well-known reports showing that the rubbing strength affects the azimuthal (apolar) surface anchoring energy [26,27] as well as the surface tension [28]. In the rubbing procedure, a rapidly rotating cloth-wrapped roller is applied to the surface of the polyimide film, with the sample stage moving under the roller generating friction on the surface of the polymer film. The rotational velocity, $A_{\text{Roller}}$ [rpm], translational velocity, $A_{\text{Table}}$ [mm/s], of the stage, and the force exerted by the cloth compression are important variables influencing the effect of rubbing. To quantitatively assess the intensity of rubbing, Uchida et al. [26] investigated the rubbing density (distance), $L$, defined as follows:

$$L = Nl\left(1 + \frac{2\pi r A_{\text{Roller}}}{60 A_{\text{Table}}}\right) \tag{1}$$

Here, $N$ represents the number of rubbing applications, $l$ [mm] is the contact length of the rubbing roller, $r$ [mm] is the roller's radius, $A_{\text{Roller}}$ [rpm] is the rotation rate of the roller, and $A_{\text{Table}}$ [mm/s] is the movement speed of the stage. In this study, $l$ is equated to the compression amount, $M$ [mm], and the rubbing intensity, $L$, is then calculated using:

$$L = 2N\sqrt{2rM - M^2}\left(1 + \frac{2\pi r A_{\text{Roller}}}{60 A_{\text{Table}}}\right) \tag{2}$$

The significance of $L$ having a length dimension should be emphasized since $L$ represents the total length of the rubbing cloth that contacts a certain point of the polyimide film surface [26]. As for the rubbing condition, $N = 1$, $r = 26.15$ mm, $M = 0.3$ mm $A_{\text{Table}} = 5$ mm/s, and $A_{\text{Roller}}$ ranges from 53 to 328 rpm. The resultant $L$ ranges from 500 to 3000 mm. As a reference, the azimuthal anchoring energy of polyimide combined with the cyanide nematic mixture ZLI-2293 (Merck) showed a gradual rubbing strength dependence on the order of $10^{-4}$ J/m$^2$ [29], and the pretilt angle was less than 2 degrees.

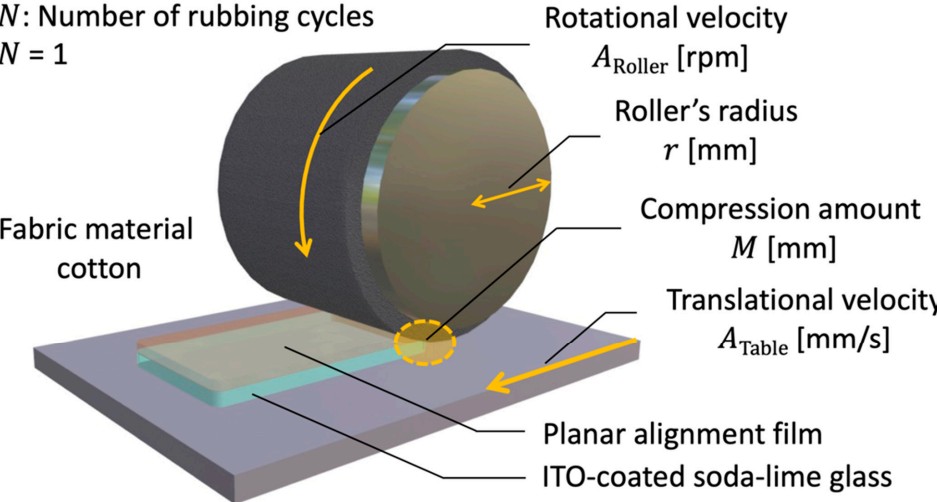

**Figure 2.** Schematic of the rubbing apparatus designed to rub the horizontally aligned LC film formed on a glass substrate with a cotton pad.

Sandwich-type cells were created by arranging two substrates with their rubbed polyimide surfaces facing inward such that their rubbing directions were anti-parallel to each other to promote planar alignment of the NLC. The gap between the pair of glass substrates (i.e., the cell thickness) was maintained using droplets of UV-curable (Photolec A, SEKISUI CHEMICAL Co., Ltd., Tokyo, Japan) and thermosetting adhesives (Photolec E, SEKISUI CHEMICAL Co., Ltd., Tokyo, Japan). The LC material was injected into the cell

through capillary action at the isotropic phase temperature. During the cooling process after the injection of the LC material, little temperature gradient between the top and bottom substrates occurred in the measurement, which contributed to maintaining the director's alignment symmetry [30,31].

The cell thickness was estimated using optical interference through POM (ECLIPSE LV100N POL, Nikon Co., Ltd., Tokyo, Japan) and optical spectrometry (BLUE-Wave, StellarNet Co., Ltd., Tampa, FL, USA) on the empty cell before LC injection. The LC texture was recorded using POM and a microscope-based USB digital camera (WRAYCAM-NF300, WRAYMER Inc., Osaka, Japan). The control software WraySpect ver. 2.7.1 was employed for capturing and editing purposes, and images were taken using the auto-white balance feature.

### 2.2. Brewster Angle Reflection (BAR) Method

As illustrated in Figure 3a, when the wavelength of the incident light is consistent with the absorption band of the dye, the propagated light is entirely absorbed by the LC bulk, prohibiting its passage through the LC cell. Hence, the reflection is uniquely derived from the front substrate, with no contribution from the bulk or the bottom substrate of the LC cell. The nature of the reflection from the surface between the alignment film and the LC is intricately influenced by the LC director alignment near the alignment surface. When the light is incident on the glass substrate at the Brewster angle, as expressed by Snell's law, *p*-polarized light undergoes unhindered transmission through the air–glass boundary without reflection. The detected light is composed of the predominant reflections from the surface, namely the glass–ITO film, ITO film–alignment film, and alignment film–LC, as depicted in Figure 3a. Investigating this reflective behavior provides insight into the LC director alignment at this juncture. Using this principle, Okutani et al. introduced the BAR method [25]. They analyzed the resultant light intensity to precisely estimate the LC director alignment at the surface. In our study, we configured the experimental apparatus, as shown in Figure 3b, to perform measurements through the BAR method.

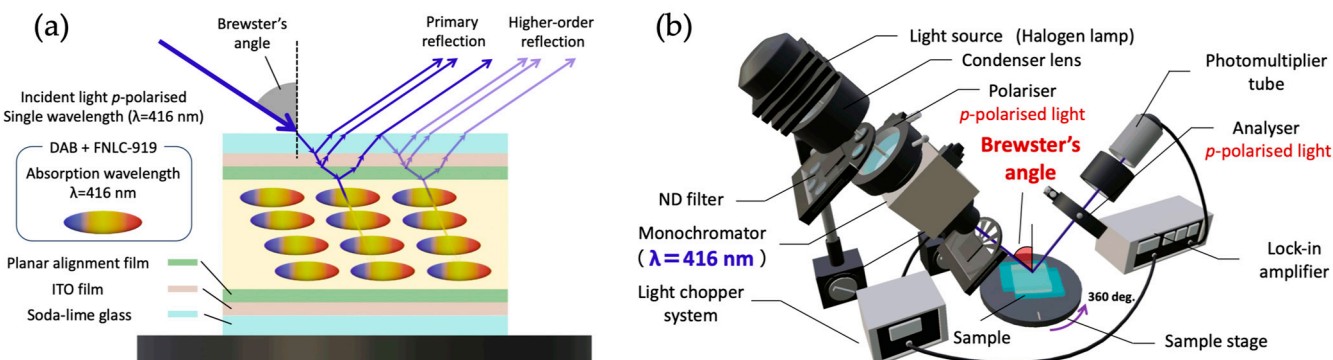

**Figure 3.** Principle of the BAR method. (**a**) The reflective characteristics of light at the Brewster angle are employed to analyze the director alignment at the LC surface. The absorption peak wavelength for the dye-added LC was observed at 416 nm. When the wavelength of the incident light falls within the absorption band of the dye, it is entirely absorbed within the LC cell, and the observed reflected light is solely from the front substrate, thus facilitating precise investigation into the director information at the surface. (**b**) Experimental setup for the BAR method. Light from a monochromator is introduced at a single wavelength. Both the incident and reflective sides are equipped with analyzer and polarizer to control the polarization. A light chopper modulates the light intensity, enabling high-sensitivity measurements by eliminating noise.

### 3. Consideration for Twist Unwind Based on the Torque Balance Equation

In applications utilizing LCs as devices, it is necessary to align the LC molecules in a specific direction. The surface anchoring effect arises from the mutual interactions between the alignment film surface and the LC molecules. It is pointed out that the surface

anchoring energy should be expressed as the sum of polar and nonpolar terms (Rapini-Papoular surface potential) especially in the case of ferroelectric liquid crystals [32–34]. When the surface anchoring energy at the alignment film is sufficiently high, the binding force for LC molecules is strong and unaffected by alignment distortions in the bulk, thus maintaining a consistent alignment state at the surface. This condition is referred to as strong anchoring. In FNLCs, the formation of a $\pi$-twist structure in the bulk under strong anchoring conditions has been confirmed when using a thick LC layer with an antiparallel rubbing manner [15]. Polar anchoring is likely to be predominant in polyimide-based alignment films [14,16]. We studied the polar alignment structure through POM to validate the phenomenon of polar anchoring energy predominance in FNLC-919 by tuning the rubbing strength and cell thickness parameters. We assumed that polar anchoring at the surface is the dominant factor in LC alignment. We derive a simplified torque balance equation from classical continuum theory and provide a phenomenological explanation of the experimental results. For this, we consider the FNLC polar alignment cell shown in Figure 4. Allowing the easy axis of the alignment of the lower substrate, $n_0$, to be in the *x*-axis direction, we define the substrate plane as the *xy* plane and the direction perpendicular to the substrate plane as the *z*-axis, designating the cell thickness as *d*. We designate the easy axis of alignment of the upper substrate as $n_d$ and its azimuthal angle as $\varphi_d$. An external electric field, *E*, is defined within the *xy* plane with the azimuthal angle $\varphi_E$, whereas no external electric field is applied in this experiment. Defining such cartesian coordinate axes in this manner, the easy axis of alignment at the lower surface and the upper surface are represented as:

$$n_0 = (1, 0, 0), \tag{3}$$

$$n_d = (\cos \varphi_d, \sin \varphi_d, 0). \tag{4}$$

The azimuthal angle of the director at the position, *z*, which is represented as $n(z)$, is denoted by $\varphi(z)$.

$$n(z) = (\cos \varphi(z), \sin \varphi(z), 0) \tag{5}$$

The spontaneous polarization, $p(z)$, is parallel to the director, $n(z)$:

$$p(z) = p\,n(z), \tag{6}$$

where *p* is the magnitude of the spontaneous polarization. In scenarios where there is in-plane alignment and the electric field points in the in-plane direction, the only alignment distortion is a twist distortion. Consequently, when calculating the elastic free energy density $f_{\text{elas}}$, it is appropriate to solely consider the influence of the twist distortion:

$$f_{\text{elas}} = \frac{1}{2} K_T \left( \frac{d\varphi}{dz} \right)^2, \tag{7}$$

where $K_T$ is the intrinsic twist elastic constant for the FNLC in bulk. The energy due to the electric field, denoted by $f_E$, is expressed as:

$$f_E = -p(z) \cdot E = -pE \cos(\varphi(z) - \varphi_E) = 0 \tag{8}$$

The bulk free energy density, $f_B$, is given by the summation of $f_{\text{elas}}$ and $f_E$. The polar anchoring energy at the lower and upper surface can be expressed from the symmetry:

$$F_S^{(0)} = -W_0 \cos(\varphi(0) - 0) = -W_0 \cos(\varphi(0)) \tag{9}$$

$$F_S^{(d)} = -W_d \cos(\varphi(d) - \varphi_d) \tag{10}$$

The total free energy, *F*, per unit area of the FNLC cell is:

$$F = \int_0^d f_B(z)\,dz + F_S^{(0)} + F_S^{(d)} \tag{11}$$

To derive the conditions under which Equation (11) is minimized, we consider a perturbation in $\varphi(z)$ resulting in a negligible change in $F$, $\delta F$. Using a Taylor series expansion for the multivariate function $f_B\left(\varphi, \frac{d\varphi}{dz}\right)$, only the first term is considered here for a succinct understanding of the local behavior. This yields:

$$\delta F = \int_0^d \left[ \frac{\partial f_B}{\partial \varphi} \delta\varphi + \frac{\partial f_B}{\partial\left(\frac{dp}{dz}\right)} \frac{d\delta\varphi}{dz} \right] dz + F_S^{(0)} + F_S^{(d)}. \tag{12}$$

Integration by parts further refines this into:

$$\int_0^d \left( \frac{\partial f_B}{\partial \varphi} - \frac{d}{dz} \frac{\partial f_B}{\partial(d\varphi/dz)} \right) \delta\varphi dz + \left( \frac{\partial F_S^{(d)}}{\partial \varphi(d)} + \left[ \frac{\partial f_B}{\partial(d\varphi/dz)} \right]_{z=d} \right) \delta\varphi(d)$$
$$+ \left( \frac{\partial F_S^{(0)}}{\partial \varphi(0)} - \left[ \frac{\partial f_B}{\partial(d\varphi/dz)} \right]_{z=0} \right) \delta\varphi(0) = 0 . \tag{13}$$

For Equation (13) to hold for any infinitesimal variation of $\varphi(z)$,

$$\text{in the bulk } \frac{d}{dz} \frac{\partial f_B}{\partial(d\varphi/dz)} - \frac{\partial f_B}{\partial \varphi} = 0, \tag{14}$$

$$\text{at the lower surface } \frac{\partial F_S^{(0)}}{\partial \varphi(0)} - \left[ \frac{\partial f_B}{\partial\left(\frac{d\varphi}{dz}\right)} \right]_{z=0} = 0, \tag{15}$$

$$\text{at the upper surface } \frac{\partial F_S^{(d)}}{\partial \varphi(d)} + \left[ \frac{\partial f_B}{\partial(d\varphi/dz)} \right]_{z=d} = 0. \tag{16}$$

For these equations to be valid for any perturbation:

$$\text{In the bulk } \frac{d^2\varphi}{dz^2} - \frac{pE}{K_T} \sin(\varphi(z) - \varphi_E) = 0 , \tag{17}$$

$$\text{in the lower surface } \left( \frac{d\varphi}{dz} \right)_{z=0} - \frac{W_0}{K_T} \sin(\varphi(0) - \varphi_0) = 0 , \tag{18}$$

$$\text{in the upper surface } \left( \frac{d\varphi}{dz} \right)_{z=d} + \frac{W_d}{K_T} \sin(\varphi(d) - \varphi_d) = 0 . \tag{19}$$

The azimuthal alignment of the LC directors, dependent on the cell thickness assumed in this analysis, is explicitly shown in Figure 4 along with the polar alignment structure model. This study uses the collocation method for numerical computations [35]. For the initial estimations, the approximation functions are defined by Equations (20) and (21):

$$\varphi_{init}(z) = \varphi(0) + z \cdot \frac{\varphi(d) - \varphi(0)}{d}, \tag{20}$$

$$\left( \frac{d\varphi}{dz} \right)_{init} = -\frac{W_0}{K_T} + z \cdot \frac{\frac{W_d}{K_T} - \left(-\frac{W_0}{K_T}\right)}{d}. \tag{21}$$

As inferred from Equation (17), the director throughout the cell linearly varies because the second term on the left side of Equation (17) is 0 when no external electric field is applied (i.e., $E = 0$). In the numerical analysis, parameters were set with a node count of 1000 and a permissible error of 0.001. The computational results are shown in Figure 5. In this analysis, the antiparallel cell where the rubbing directions between the top and bottom substrates are antiparallel is supposed. The elastic constant, $K_T$, is taken to be 5.0 pN. The estimated polar anchoring energy, $W_{0,d}$, is approximately $1.0 \times 10^{-5}$ J/m$^2$ [3,16] where $W_{0,d}$ (i.e., $W_0$ and $W_d$) are assumed to be equal. It should be noted that we cannot assume

that a decrease in the cell thickness, $d$, decreases the genuine surface polar anchoring energy, $W_{0,\text{d}}$. Instead, $W_{0,\text{d}}$ should be understood through the extrapolation length or the reduced anchoring energy, $W_n$, defined by:

$$W_n = \frac{W_0 d}{K_{\text{T}}},\qquad(22)$$

which is easily derived from Equation (17) or (18). The intrinsic twist is determined to be decreased due to the decrement of $W_n$ and the increment of $\varphi_{\text{d}}$. Though the intermolecular interactions of the bulk polar molecules remain constant, the twist angle (i.e., $\varphi_{\text{d}} - \varphi_0$) is found to be dependent on the cell thickness, $d$, as long as $K_{\text{T}}$ and $W_0$ do not change. This is the very essence of torque balance.

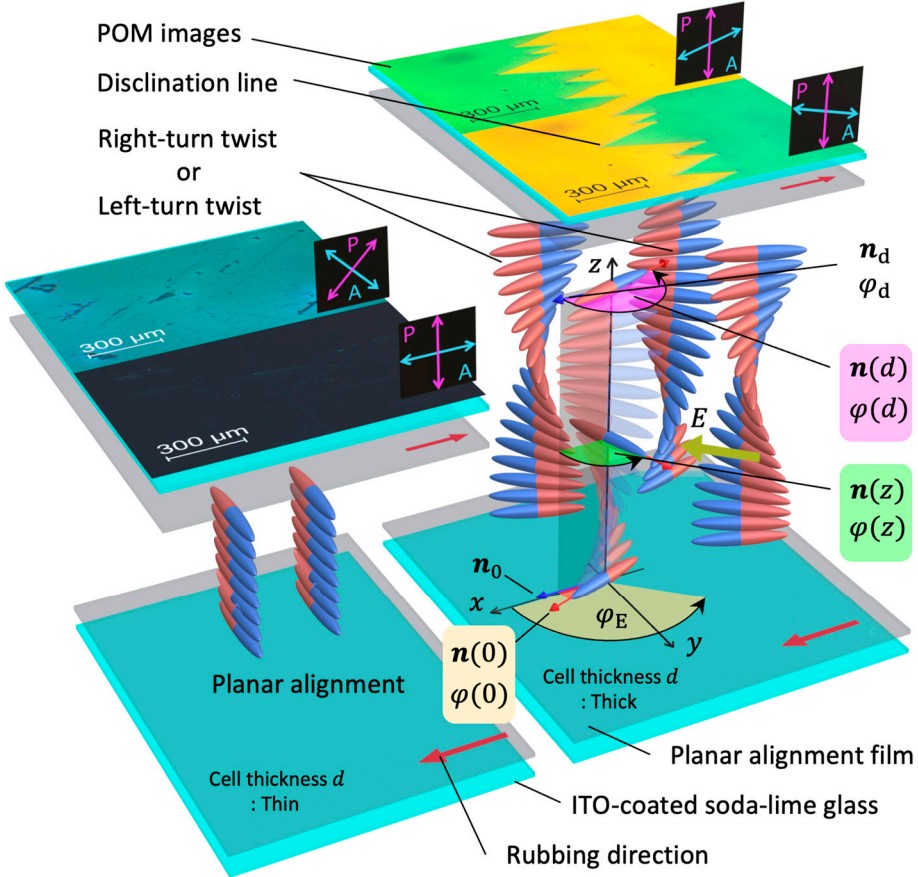

**Figure 4.** Polar alignment structure model of FNLCs derived from theoretical analysis. The POM results from Figure 6 are shown for both thin and thick cell thickness. The in-plane tilt angle and the twist sense are properly defined.

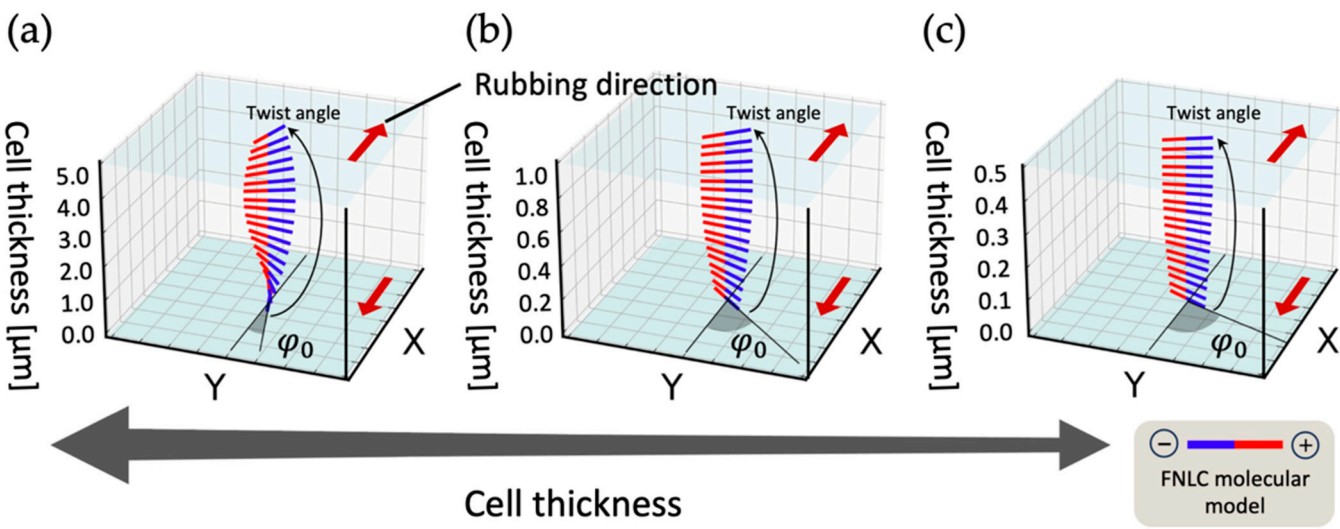

**Figure 5.** Numerical analysis results of the alignment structure theoretical model in antiparallel cells of FNLC under three cell thickness conditions of (**a**) 5.0 μm, (**b**) 1.0 μm, and (**c**) 0.5 μm. This analysis is based on Equation (17) and the boundary conditions (18) and (19), and the resultant $\varphi_0$ and twist angles are (**a**) 15.1° and 149.8°, (**b**) 47.2° and 85.5°, and (**c**) 63.5° and 53.0°.

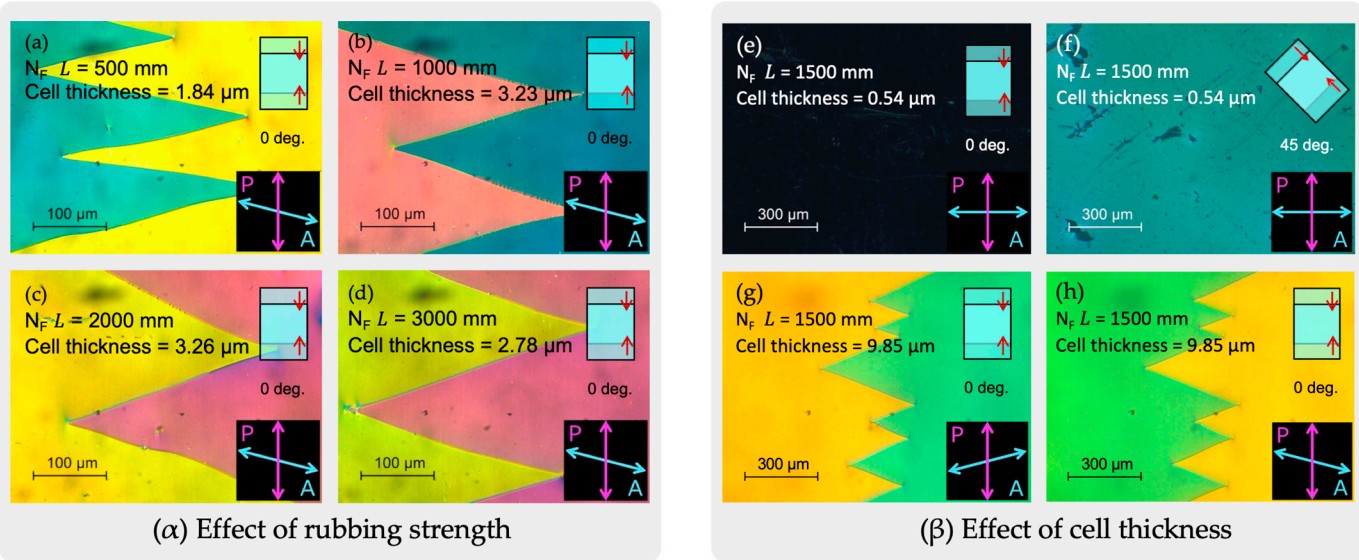

**Figure 6.** Texture of an FNLC photographed using POM under crossed Nicol. The illustration inside each photograph represents the cell's angle relative to the analyzer (A) and the polarizer (P). The angle between the analyzer and polarizer was slightly shifted to reveal the difference in the twist sense in the π-twist structure. Small red arrows indicate the rubbing direction. (**α**) Relationship between rubbing strength and FNLC polar alignment structure. The rubbing strength, *L*, was (**a**) 500 mm, (**b**) 1000 mm, (**c**) 2000 mm, and (**d**) 3000 mm, respectively. From these photographs, typical π walls were found regardless of rubbing strength. (**β**) Cell thickness dependence. The rubbing strength, *L*, was set 1500 mm. Cell thickness was (**e**,**f**) 0.54 μm (**g**,**h**) 9.85 μm, respectively. From these photographs, a uniform alignment was observed in the case of the thin cell, whereas a typical π-twist texture was observed in the case of the thick cell.

## 4. Experimental Results and Discussion

### 4.1. Rubbing Strength and Polar Alignment Structure

While the influence of the rubbing strength on surface azimuthal anchoring has been unraveled in past research for ordinary NLCs [27], the effect on polar alignment in polar molecules, such as FNLCs, is not comprehensively understood. When the polar anchoring energy dominates at the surface, it is expected that variations in rubbing strength will not affect the bulk polar alignment structure. To validate this hypothesis, the rubbing strength, $L$, was varied from 500 to 3000 mm, and the changes in the polar alignment structure throughout the cell were observed using POM. Figure 6 shows textures of the FNLC photographed using POM under crossed Nicol. The illustration inside each photograph represents the cell's angle relative to the analyzer (A) and the polarizer (P). The angle between the analyzer and polarizer is slightly shifted to reveal the difference in the twist sense in the $\pi$-twist structure. Small red arrows indicate the rubbing direction. On the one hand, as shown in Figure 6 ($\alpha$), typical $\pi$-walls (zig-zag line) were found regardless of rubbing strength when the cell thickness was thicker than 1 $\mu$m. On the other hand, as shown in Figure 6 ($\beta$), a uniform alignment was observed when the cell thickness was thinner than 1 $\mu$m (e,f) since a clear extinction position can be confirmed. These experimental observations show that the rubbing strength is not a critical determinant for the polar alignment structure. The texture difference in Figure 6 ($\beta$) is interpreted as indicating that the twist structure unwinds as the cell thickness becomes thinner and the reduced surface polar anchoring energy, $W_n$, falls below a certain value. This interpretation is also quantitatively plausible from a comparison with Figure 5c.

### 4.2. Director near the Alignment Surface (Easy Axis)

Under the theoretical framework described in the previous section, the director of the FNLC molecules at the surface (the so-called easy axis) tends to align perpendicular to the rubbing direction when $W_n$ is below a certain value. Unfortunately, the direction of the easy axis cannot be determined only from Figure 6. The BAR method was introduced to overcome this limitation, enabling an estimation of the easy axis direction. Figure 7 shows photographs using POM under crossed Nicol and the polar plots of the reflection light intensity (arbitrary unit) for cells whose cell thickness were of 2.28, 1.12, and 0.68 $\mu$m, respectively. The textures showing the twisted structures of LC molecules in Figure 7a,b, which are intrinsic to the polar alignment structure of FNLC, indicate that $N_F$ is preserved despite the DAB addition. In the BAR measurements, the reflection light intensities from both the top substrate (blue dots) and the bottom substrate (red dots) of the cell were assessed by flipping the cell over on the sample stage depicted in Figures 3b and 7c,f,i where they represent polar plots of the reflection light intensities when rotated in-plane around the normal to the sample cell. The rubbing direction was set to $0°$ in the polar plots for both substrates. Note that even the BAR method cannot distinguish between the head and tail of a molecule. For a cell thickness of 2.28 $\mu$m, as shown in Figure 7c, two reflectance peaks were observed in the direction perpendicular to the rubbing direction. This is because the angle at which the reflectance peak appears is determined by the angle of the easy axis when the difference between the refractive index of the substrate and the effective refractive index of the FNLC-919 is maximum. In this series of experiments, it is reasonable to interpret that the direction perpendicular to the easy axis coincides with that of maximum reflectance because it is reasonable for the easy axis in the $\pi$-twist alignment to be oriented in the rubbing direction. For a cell thickness of 1.12 $\mu$m, which is different from the former case, the shape of the polar plot of the reflectance measured on the front surface is not similar to that measured on the back surface. This result implies that the easy axis does not coincide with the rubbing direction when the cell thickness is thin. When the cell thickness is 0.68 $\mu$m, it seems that the easy axis could not be determined from the polar plot of the reflectance, while a uniform orientation is obtained from Figure 7g,h. Here, to determine the direction of the bulk director of the unwind cell, an optical retarder was inserted between the cell and analyzer under the crossed Nicol, and the directors in the bulk

and the rubbing direction were found to be parallel [23]. This means that the assumption as such that the $W_0$ and $W_d$ are equal is not appropriate, and one side of the polar anchor is lost as pointed out by Chen et al. [15]. These experimental results suggest that the direction of the easy axis depends on the cell thickness. The consistency between the numerical analysis and the experimental results supports the validity of the proposed model.

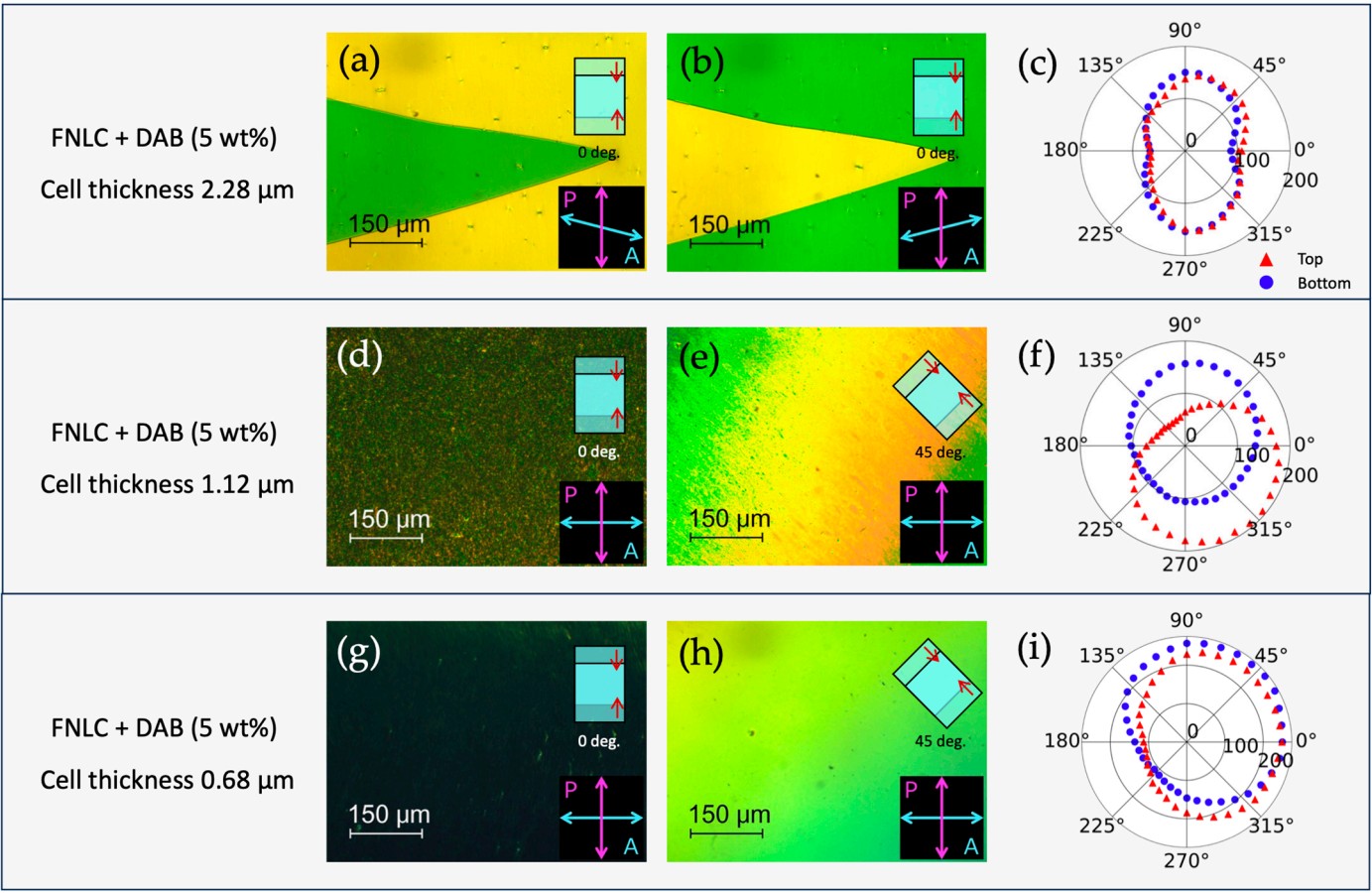

**Figure 7.** Texture of the FNLC photographed using POM under crossed Nicol and the BAR measurements for the cell thicknesses of 2.28, 1.12, and 0.68 μm filled with FNLC-919 doped with 5 wt% DAB. The illustration inside each photograph (**a**,**b**,**d**,**e**,**g**,**h**) represents the cell's angle relative to the analyzer (A) and the polarizer (P). Small red arrows indicate the rubbing direction. (**c**,**f**,**i**) represent the polar plots of the reflection light intensities (arbitrary unit). In the BAR measurements, the reflection light intensities from both the top substrate (blue dots) and the bottom substrate (red dots) of the cell were assessed by flipping the cell over on the sample stage depicted in Figure 3b. The rubbing direction was set to 0° in the polar plots for both substrates.

## 5. Conclusions

Research on the characteristic alignment of FNLCs, such as the twist and conic structure, has been vigorously carried out on DIO and RM734 through which the FNLC phase was first discovered. However, reproducibility should also be confirmed on other FNLCs that exhibit the FNLC phase at room temperature. We came up with the concept that the classical continuum theory is applicable to estimating the polar anchoring, which is an essential phenomenon in FNLC. The analytical formula based on the simplified torque balance equation predicts that the twist angle in the FNLC bulk decreases while thinning the cell thickness. We numerically showed that its essence can be explained through the concept of reduced polar anchoring energy. To confirm this concept's validity, we performed the BAR method and POM observation and, as a result, qualitative agreement was obtained. In particular, the BAR method provides experimental evidence that the

director of FNLC molecules at the surface changes with cell thickness, as shown in our numerical analysis.

**Author Contributions:** Conceptualization, S.A. and M.K.; methodology, S.A. and M.K.; software, S.A.; validation, S.A., M.K. and T.A.; formal analysis, M.K.; investigation, S.A.; resources, M.K.; data curation, S.A.; writing—original draft preparation, S.A.; writing—review and editing, Y.S. and M.K.; visualization, S.A.; supervision, M.K.; project administration, M.K.; funding acquisition, M.K. All authors have read and agreed to the published version of the manuscript.

**Funding:** This study was partly supported by the Japan Society for the Promotion of Science (JSPS) KAKENHI Grants 23K03935.

**Data Availability Statement:** The data and program code that support the findings of this study are openly available in our laboratory's homepage at https://alcllan.nagaokaut.ac.jp/kimura/research/Crystals_raw_data2024.zip.

**Acknowledgments:** We sincerely thank Merck Electronics KGaA, Darmstadt, Germany, and JSR Co., Ltd., Tokyo, Japan, for providing us with the LC and polyimide materials. We also wish to thank Professor Seiichi Kawahara of Nagaoka University of Technology for their collaboration in the DSC measurements.

**Conflicts of Interest:** There are no conflicts to declare.

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
