# Peer review of "Self-Consistent Explanation of the Untwist Alignment of Ferroelectric Nematic Liquid Crystals with Decreasing Cell Thickness and Deviation of the Surface Easy Axis Experimented upon Using the Brewster Angle Reflection Method"

_crystals, doi:10.3390/cryst14020157_

Round 1
Reviewer 1 Report
Comments and Suggestions for Authors
Dear Authors,
The study of ferroelectric nematic LC alignment covers an interesting topic that remains important to expert society. Due to spontaneous polarization, these LC materials manifest specific alignment properties.
I propose that the authors study the following paper on alignment layers' role in FLC alignment [S. P. Palto et al. The new role of alignment layers in bistable switching of ferroelectric liquid crystals. Numerical simulation and experimental results. Ferroelectrics, Vol. 310, p.95-109 (2004). https://doi.org/10.1080/00150190490510456]. There you might find useful interpretation of a model for anchoring strength influence on the favorable alignment arrangement. It could be cited in section 4.1.
Please, add (a, b, с) labels in Fig.5.
I’m not very good at judging theoretical models. As to the experiment shown in Fig.7, possibly the 7i could be showing that there is a pretilt of the molecules to the cell plane. The exact retardation of the LC cell could be determined using the Bertrand lens in POM and compared to the calculated retardation using known delta n and cell thickness. As the pretilt reduces the layer retardation, you will be able to estimate its input.
Comments on the Quality of English LanguageAs to English,
very huge -> huge.
Line 285 – note -> noted.
Reviewer 2 Report
Comments and Suggestions for Authors
I have thoroughly reviewed the manuscript by Abe et al., which addresses an important aspect in the field, exploring weak polar anchoring effects in anti-parallel rubbed planar cells of ferroelectric nematic liquid crystals, which are hot topic nowadays. The authors report on the deviation of the director from the rubbing direction at the alignment layers in case of weak rubbing or in thin cells. The manuscript presents experimental findings supported by numerical simulations based on a continuum theoretical calculation. In my opinion, this work makes a significant and novel contribution to the understanding of molecular alignment in ferroelectric nematic liquid crystals, combining theoretical modeling with experimental validation effectively. I recommend the acceptance of this paper for publication after a revision.
Please find my comments/questions below:
1. Please fix Figure 2, since there are some rendering errors on it appearing as probably unwanted characters.
2. Please check Fig. 4. I see there letters like d, (d), (z) etc. that probably were not meant to be those.
3. In Figure 6, the exclamation marks should probably be L -s.
4. In section 3, the nomenclature is not consistent as regards the z-dependence argument of the director (and the polarization): n(z). Sometimes (z) is in lower index, sometimes not. Also, I think z should not be bold as it is not a vector.
5. In the theoretical model, an external electric field is considered first, then it is set to zero. The ferroelectric material has spontaneous polarization even without external electric field, therefore it can generate a depolarizing electric field, which can be high. As regards the director configuration and the weak anchoring effects, wouldn’t the effect of depolarizing electric field be important?
6. I think the authors must discuss the optical effect of getting outside of the Mauguin limit in case of small thicknesses. In case of light transmission, for low thicknesses, the light polarization cannot follow the director twist, and optically, the cell behaves as it would be a birefringent plate with the axis of the average direction of the twisted structure. For a pi-twist, the average is exactly pi/2. In this sense, even an intact pi-twisted structure can appear to be as it would be homogeneous director at pi/2 direction. This may change the conclusion of the existence of a weak anchoring effect. The best would be to simulate the optical transmission by e.g. Jones matrix method.
7. As regards the reflection from thin cells, it is questionable what is the effective direction shown by the BAR experiments. The authors write in the manuscript that the penetration depth of the used light is about 1 micron. If we assume a perfect pi-twisted structure at low thickness, the director gradient is quite large. Thicknesses of about a half micron is comparable to the light wavelength. The authors must discuss and clarify the interpretation of the BAR experiments when the optic axis may rotate pi within a distance of about the same as one wavelength of light. The question is whether the apparent rotation at the anchoring surface is originated in a weak anchoring effect or it is an artefact due to the reflection from a very thin twisted structure.
8. In order to increase the visibility of their paper, optionally, the authors may consider to complete the introduction by referring to some recent papers about the same room temperature mixture FNLC-919: https://doi.org/10.1002/adfm.202314158 and https://doi.org/10.1002/advs.202305950.
Round 2
Reviewer 2 Report
Comments and Suggestions for Authors
The manuscript by Abe et al. has been sufficiently improved compared to its previous state, therefore I recommend the publication. Nevertheless, Figures 2, 4, and 6 still have rendering problems that must be fixed. I checked with Adobe Acrobat and SumatraPDF viewers showing the same errors in the pdf file.